

# Hydrometeorological data from Marmot Creek Research Basin, Canadian Rockies

Xing Fang[1], John W. Pomeroy[1], Chris M. DeBeer[1], Phillip Harder[1], and Evan Siemens[1]

[1]Centre for Hydrology and Global Institute for Water Security, University of Saskatchewan, Saskatoon, S7N 1K2, Canada

*Correspondence to*: Xing Fang (xing.fang@usask.ca)

**Abstract.** Meteorological, snow survey, streamflow, and groundwater data are presented from Marmot Creek Research Basin, Alberta, Canada. The basin is a 9.4 km$^2$, alpine-montane forest headwater catchment of the Saskatchewan River Basin that provides vital water supplies to the Prairie Provinces of Canada. It was heavily instrumented, experimented upon and operated by several federal government agencies between 1962 and 1986, during which time its main and sub-basin streams were

gauged, automated meteorological stations at multiple elevations were installed, groundwater observation wells were dug and automated, and frequent manual measurements of snow accumulation and ablation and other weather and water variables were made. Over this period, mature evergreen forests were harvested in two sub-basins, leaving large clear-cuts in one basin and a "honeycomb" of small forest clearings in another basin. Whilst meteorological measurements and sub-basin streamflow discharge weirs in the basin were removed in the late 1980s, the federal government maintained the outlet streamflow discharge

measurements and a nearby high elevation meteorological station, and the Alberta provincial government maintained observation wells and a nearby fire weather station. Marmot Creek Research Basin was intensively re-instrumented with 12 automated meteorological stations, four sub-basin hydrometric sites and seven snow survey transects starting in 2004 by the University of Saskatchewan Centre for Hydrology. The observations provide detailed information on meteorology, precipitation, soil moisture, snowpack, streamflow, and groundwater during the historical period from 1962 to 1987 and the

modern period from 2005 to the present time. These data are ideal for monitoring climate change, developing hydrological process understanding, evaluating process algorithms and hydrological, cryospheric or atmospheric models, and examining the response of basin hydrological cycling to changes in climate, extreme weather, and land cover through hydrological modelling and statistical analyses. The data presented are publicly available from Federated Research Data Repository (https://dx.doi.org/10.20383/101.09).

# 1 Introduction

The eastern slopes of the Canadian Rocky Mountains form the headwaters of the Saskatchewan River Basin (SRB), whose water supplies are vital to domestic, agricultural, and industrial users in the Canadian Prairie Provinces. These mountain headwaters occupy about 12.6% of total drainage area but generate 87% of total water yield in the SRB (Redmond, 1964). Recognising the importance of these headwaters, the Eastern Rocky Mountain Forest Conservation Act was passed in 1947,



which aimed to conserve and protect the Saskatchewan River headwaters (Neill, 1980; Rothwell et al., 2016). The Eastern Slopes (Alberta) Watershed Research Program (AWRP) was created in 1960 to investigate relationships between forest, soil, climate, and water and to examine the impacts of commercial timber harvesting practices on basin water yield and water quality (Jeffery, 1965; Kirby and Ogilvy, 1969). This program was a collaborative effort between several provincial and federal government agencies to establish experimental watersheds in the headwaters, one of which was the establishment of what was then called the "Marmot Creek Experimental Watershed" during 1961-1962 (Rothwell et al., 2016), this later became the University of Saskatchewan-operated "Marmot Creek Research Basin" (MCRB) by which it is referred to in this paper.

During the historical period of 1962-1986, a paired-basin experiment devised by the Canadian Forestry Service (CFS) explored the effects of forest cutting on snow accumulation and water yield in MCRB. Two types of forest clearing experiment were conducted in the sub-alpine spruce/fir forest part of basin: six large "commercial" forest cut blocks were harvested in the Cabin Creek sub-basin during 1971-1972 and a "honeycomb" of numerous small circular clearings, each 12 m to 18 m in diameter, were harvested in the Twin Creek sub-basin during 1977-1979, with Middle Creek left intact as a control sub-basin (Rothwell et al., 2016). Snow accumulation increased by 21% in the large forest cutting blocks (Swanson et al., 1986), and 28% in the small forest clearings compared to under adjacent intact forest canopies (Swanson and Golding, 1982). Overall, there was no statistically significant change in streamflow that could be associated with the forestry manipulations (Harder et al., 2015). Several other studies were carried out in parallel to the forest clearing experiments. Investigations on soil water storage and soil temperature in relation to snow accumulation and melt, forest, and slope orientation were conducted at several sites in MCRB and provided some early understanding of infiltration and runoff in the basin (Harlan, 1969; Hillman and Golding, 1981). Extensive field campaigns throughout MCRB produced detailed descriptions of soils (Beke, 1969) and surficial geology (Stevenson, 1967). Additional studies were undertaken to assess the basin's meteorology (Munn and Storr, 1967; Storr, 1967, 1973). Most hydrometeorological observations in MCRB ceased after 1986 due to opening of adjacent Nakiska Ski Resort in the 1986-1987 ski season and subsequent hosting of 1988 Winter Olympic Games; only streamflow measurements at the main outlet by Environment and Climate Change Canada (ECCC), and groundwater measurements by AEP, were continued, though a high elevation weather station was established on Centennial Ridge by ECCC and Alberta Agriculture and Forestry maintained a nearby valley bottom weather station (Rothwell et al., 2016).

After the Olympics, research activities in MCRB were minimal until 2004 when the research basin was reactivated by the University of Saskatchewan with the help of the University of Calgary, and ECCC. Wide-ranging research has been conducted since then to improve the understanding of the impact of forest canopy and forest clearings on snow accumulation and snowmelt energetics (Ellis and Pomeroy, 2007; Essery et al., 2008; Pomeroy et al., 2009; Ellis et al., 2013; Musselman and Pomeroy, 2017), slope and aspect controls on snow accumulation and melt (DeBeer and Pomeroy, 2009; Ellis et al., 2011; Marsh et al., 2012), blowing snow and sublimation in the alpine treeline environment with respect to local wind and topography (MacDonald et al., 2010), alpine snowmelt runoff generation (DeBeer and Pomeroy, 2010), hillslope hydrology of the forest organic layer (Keith et al., 2010), and precipitation phase partitioning (Harder and Pomeroy, 2013). MCRB has also been the site of instrument or methodology development, from an early airborne LiDAR snow depth measurement (Hopkinson et al.,



2008) to acoustic measurements of snow (Kinar and Pomeroy, 2009) as well as early telescope based snow surveys (Kinar and Pomeroy, 2015). Utilizing the Cold Regions Hydrological Modelling platform (CRHM) these advances have been synthesised into a physically based hydrological model of MCRB (Fang et al., 2013), which was used to assess the impact of forest disturbances on basin hydrology (Pomeroy et al., 2012), analyse antecedent conditions on flood generation (Fang and Pomeroy,

2016) and diagnose rain-on-snow runoff generation for alpine environment during the 2013 flood in MCRB (Pomeroy et al., 2016).

    This paper includes datasets of meteorological, snow survey, streamflow, and groundwater observations measured in MCRB. Meteorological datasets include historical observations by the CFS and ECCC and recent measurements by the University of Saskatchewan Centre for Hydrology. Continuous records of streamflow measurements by ECCC and University of

Saskatchewan as well as groundwater levels monitored by AEP are also included. The snow survey data presented were conducted in clearings, under forest canopies and on hillslopes at various elevations and are useful for model evaluation and snow process studies. Some of the studies utilising these datasets document the basin resilience to changes in climate, extreme weather, and land cover (Harder et al., 2015), a sensitivity analysis of climate warming on snow processes (Pomeroy et al, 2015), and assesses variability of climate and its impact on the hydrological processes (Siemens, 2016).

## 15  2 Site description

Marmot Creek Research Basin (MCRB) (50.95°N, 115.15°W) is in the headwaters of the Bow River Basin in the Front Ranges of the Canadian Rocky Mountains (Fig. 1) and its streamflow discharges into the Kananaskis River. The basin area (9.4 km$^2$) is defined by the Water Survey of Canada stream gauge that was installed in 1962 (Bruce and Clark, 1965). MCRB is composed of three upper sub-basins: Cabin Creek (2.35 km$^2$), Middle Creek (2.94 km$^2$), and Twin Creek (2.79 km$^2$), which

converge into the Confluence Sub-basin above the main stream gauge (1.32 km$^2$). Upper Marmot Creek is an upper sub-basin of Middle Creek (1.178 km$^2$) is primarily alpine and is also gauged. Based on a 2008 LiDAR 8m digital elevation model (DEM) (Hopkinson et al., 2012), hypsometric curves were derived for MCRB and its three sub-basins (Fig. 2). Elevation ranges from 1590 m a.s.l. (above sea level) at the main Marmot Creek gauging station to 2829 m at the summit of Mount Allan. Most of MCRB is covered by needleleaf vegetation which is dominated by Engelmann spruce (*Picea engelmanni*) and

subalpine fir (*Abies lasiocarpa*) in upper-mid elevations of basin (1710 to 2277 m). The lower elevation (1590 to 2015 m) forests are mainly Engelmann spruce and lodgepole pine (*Pinus contorta* var. Latifolia) with trembling aspen (*Populus tremuloides)* present near the basin outlet (Kirby and Ogilvy, 1969). Alpine larch (*Larix lyallii*) and short shrubs are present around the treeline at approximately 2016 to 2379 m. Exposed rock surfaces, grasses and talus are present in the highest alpine part of basin (1956 to 2829 m). Physiographic descriptions of these ecozones are shown in Table 1 and they are mapped in

Figure 1. These ecozones were determined from the forest cover map by the Alberta Forest Service (1963) with recent updates from site visits. Forest management experiments conducted in the 1970s and 1980s left six large clear-cut blocks (1838 to 2062 m) in the Cabin Creek sub-basin and numerous small circular forest clearings (1762 to 2209 m) in the Twin Creek sub-



basin (Golding and Swanson, 1986). The surficial soils are primarily poorly developed mountain soils consisting of glaciofluvial, surficial till and postglacial colluvium deposits (Beke, 1969). Relatively impermeable bedrock is found at the higher elevations, whilst the rest of basin is covered by a deep layer of coarse and permeable soil allowing for rapid rainfall infiltration to subsurface layers overlying relatively impermeable shale (Jeffrey, 1965). Continental air masses control the

weather in the region, which has long and cold winters and cool and wet springs with a late spring/early summer precipitation maximum. Westerly warm and dry Chinook (foehn) winds lead to brief periods when the air temperature exceeds $0\,^{\circ}$C during the winter months – these events can result in snowpack ablation at lower elevations. Annual precipitation ranges from 600 mm at lower elevations to more than 1100 mm at the higher elevations, of which approximately 70 to 75% occurs as snowfall with the percentage increasing with elevation (Storr, 1967). Mean monthly air temperature ranges from 14 $^{\circ}$C observed at

1850 m in July to -10 $^{\circ}$C observed at 2450 m in January. Mean air temperatures have increased by 2.3 $^{\circ}$C from 1967 to 2013, but there are no trends in precipitation or streamflow (Harder et al., 2015).

## 3 Meteorological data

### 3.1 Recent quality controlled data

Quality controlled (QC) 15-minute interval hydrometeorological data were processed from raw data measured at the recent

stations in MCRB: Hay Meadow, Level Forest, Upper Clearing, Upper Clearing Tower, Upper Forest, Vista View, Fisera Ridge, and Centennial Ridge. Photos of these stations are shown in Fig. 3, and Table 2 shows a list of the variables in the QC data along with instrumentation, record length and location for the stations. Most current stations started measurements in 2005 and cover 11 water years (WY) from 1 October 2005 to 30 September 2016 (WY2006 to WY 2016) with two exceptions: Upper Clearing Tower and Fisera Ridge; the former started data collection 21 October 2007 and the latter started data collection

13 October 2006. The QC data were generated by applying a quality assurance procedure to remove erroneous data in the 15-minute raw data. Table 3 lists the QC thresholds used to remove: 1) measurements outside of defined maximum and minimum ranges; 2) measurements that exceed a rate of change (ROC) limit; 3) constant measurements due to sensor failure. In the QC data, values of -9999 denote the measurements removed from the raw data. In addition, daily QC soil moisture is provided for 11 water years from the Level Forest station and eight water years (WY2006 to WY2013) from the Upper Forest. From 19

October 2012, soil moisture is monitored at a 15-minute interval at Upper Forest and this higher temporal resolution data is included.

### 3.2 Recent modelling data

Hourly modelling data were obtained by averaging the 15-minute QC observations of air temperature ($^{\circ}$C), relative humidity (%), wind speed (m s$^{-1}$), incoming solar radiation (W m$^{-2}$), and soil temperature at either 5 cm or 10 cm below ground surface

($^{\circ}$C) and by summing the 15-minute QC observation of precipitation (mm). Missing observations of air temperature, relative humidity, wind speed, incoming solar radiation, and soil temperature were filled using either temporal averaging interpolation



or linear regression to nearby stations. When intervals of missing data were less than three hours, temporal averaging was employed where the observations of the variable three hours before and three hours after the missing interval from the same station were used to calculate the average. When the missing data interval was longer than three hours, linear regressions were developed amongst stations using the raw data, the regressions were ranked based on $r^2$ value, the regression relationship with

the highest $r^2$ value was selected to fill in the missing data. For missing precipitation, observations from nearby station were used along with seasonal precipitation adjustments for elevation to fill in the missing precipitation. The hourly modelling data are provided for 11 water years from 1 October 2005 to 30 September 2016. As described in the previous section, both Fisera Ridge and Upper Clearing Tower stations were established after WY2006, and the hourly modelling data before station establishment were estimated. For the Fisera Ridge station, air temperature, relative humidity, wind speed, incoming solar

radiation, and soil temperature from 1 October 2005 to 13 October 2006 were estimated based on linear interpolation to nearby stations, and precipitation from 1 October 2005 to 16 September 2008 was estimated from Upper Clearing precipitation with seasonal precipitation adjustments for elevation. For the Upper Clearing Tower station, the hourly incoming solar radiation measured at 20 m above ground is provided, and from 1 October 2005 to 21 October 2007 it was estimated from incoming solar radiation measured at the lower level Upper Clearing tripod station based on a linear regression because of location of

both stations in the same forest clearing. Figures 4-8 show the annual mean daily air temperature, relative humidity, wind speed, incoming solar radiation, and accumulated rainfall and snowfall with their inter-annual variability for MCRB stations for the 11 water years.

### 3.2.1 Air temperature and relative humidity

Air temperature and relative humidity were measured using Vaisala hygrothermometers at all seven stations. Table 4 shows

that average air temperature at MCRB for the 11 water years ranges from -1.6 °C at the Centennial Ridge station to -0.4 °C at the Fisera Ridge station. Both stations are located on alpine ridgetops, above treeline. Higher temperatures are found at lower elevations, where the 11-year average air temperature is 1.4 °C and 3.1 °C for the Upper Clearing station in a montane forest and the Hay Meadow station in the valley floor, respectively. WY2016 was the warmest, with the average water year air temperature being -0.3 °C, 1.0 °C, 2.7 °C, and 4.4 °C for Centennial Ridge, Fisera Ridge, Upper Clearing, and Hay Meadow

stations, respectively. WY2008 was the coolest for the Centennial Ridge and Fisera Ridge stations, with average air temperatures of -2.7 °C and -1.7 °C, respectively; whereas WY2011 was the coolest for Upper Clearing and Hay Meadow stations, with average air temperatures of 0.4 °C and 1.9 °C for Upper Clearing and Hay Meadow stations, respectively. An example of hourly air temperature and relatively humidity from Fisera Ridge station is shown in Fig. 9a and b.

### 3.2.2 Wind speed

Wind speeds were measured at all seven stations using propeller-type RM Young anemometers. The 11-water year average wind speeds on wind-exposed alpine ridges are 5.8 m s⁻¹ and 2.5 m s⁻¹ at Centennial Ridge measured at 2.41 m a.g.s. (above ground surface) and Fisera Ridge (2.55 m a.g.s.) stations, respectively. Hay Meadow, located in an open grassland valley floor

(7 m a.g.s.) has an 11-water year average wind speed of 2.0 m s$^{-1}$. Vista View station (4.11 m a.g.s.) is located in a large forest cut block with a short sparse forest cover of young trees and has an 11-water year average wind speed of 1.1 m s$^{-1}$. For the wind-sheltered stations (Upper Clearing measured at 2.85 m a.g.s, Upper Forest measured at 2.77 m a.g.s, and Level Forest measured at 2.45 m a.g.s), the 11-water year average wind speeds range from 0.1 to 0.6 m s$^{-1}$. The maximum hourly wind

speed recorded during 11 water years is 37.9 m s$^{-1}$ from Centennial Ridge station. An example of hourly wind speed from Fisera Ridge station is shown in Fig. 9c.

### 3.2.3 Incoming solar radiation

Incoming solar radiation was measured at all seven stations using Kipp and Zonen pyranometers and is included in the hourly modelling dataset except for the Vista View station due to the length of measurement. For the Upper Clearing site, hourly

incoming solar radiation measured at the top of the 20m tower station is provided in addition to that from the main tripod station near the ground (1.95 m). For the sub-canopy measurements at Upper Forest (i.e. mature spruce forest) and Level Forest (i.e. mature lodgepole forest) stations, the 11-water year mean values range from 15.9 W m$^{-2}$ (Upper Forest) to 23.7 W m$^{-2}$ (Level Forest). For the stations not affected by forest canopy, the 11-water year mean value ranges from 140.1 W m$^{-2}$ (Upper Clearing 20m tower) to 150.3 W m$^{-2}$ (Fisera Ridge). An example of hourly incoming solar radiation from the Fisera

Ridge station is shown in Fig. 9d.

### 3.2.4 Soil temperature

Soil temperature was measured using thermistors at all seven stations at either 5 cm or 10 cm below ground surface. The 11-water year mean value ranges from -0.7 °C (Centennial Ridge) to 6.5 °C (Hay Meadow). The maximum hourly soil temperature during 11 water years was 36.6 °C at the Hay Meadow station and the minimum hourly soil temperature during

11 water years was -16.5 °C at the Centennial Ridge station. An example of hourly soil temperature from Fisera Ridge station is shown in Fig. 9e.

### 3.2.5 Precipitation

Precipitation was measured with Alter-shielded Geonor T200B weighing precipitation gauges at Hay Meadow, Upper Clearing, and Fisera Ridge stations, and it was corrected for wind-induced undercatch for the wind-exposed Fisera Ridge and

Hay Meadow stations (Smith, 2007). Precipitation is divided into rainfall and snowfall based on the psychrometric energy balance precipitation phase determination method developed by Harder and Pomeroy (2013). Table 4 shows that the average annual precipitation for the 11 water years is 627 mm (229 mm snow), 839 mm (443 mm snow), and 1190 mm (802 mm snow) for Hay Meadow, Upper Clearing, and Fisera Ridge, respectively. The highest annual precipitation during the 11 water years from Fisera Ridge station was 1329 mm in WY2013 when approximately 250 mm of rainfall and snowfall fell during the June

2013 flood (Pomeroy et al., 2016), which also produced the highest annual rainfall (535 mm) recorded during the 11 water



years. An example of hourly cumulative precipitation, divided into rainfall and snowfall from Fisera Ridge station, is shown in Fig. 9f.

### 3.3 Historical modelling data

Historical meteorological data is available from the three sites shown in Fig. 1. Observations from Confluence 5 (Con 5, 50.960°N, 115.171°W, 1770 m), Cabin 5 (50.975°N, 115.182°W, 2051 m), and Twin 1 (50.957°N, 115.204°W, 2285 m) are provided. These sites were established in early 1960s by the CFS and ECCC. Based on the availability of data, continuous records of hourly air temperature (°C), relative humidity (%), and wind speed (m s$^{-1}$) and daily precipitation (mm) are included for 18 water years from 1 October 1969 to 30 September 1987. Air temperature and relative humidity were measured by thermographs or hygrothermographs (Munn and Storr, 1967); wind speed was measured by MSC type 45B anemometer, and for precipitation, Leupold-Stevens Q12M weighing gauges and MSC (Meteorological Service of Canada) tipping bucket gauges were used to take measurements for snowfall and rainfall, respectively (Storr, 1973). Data quality assurance was undertaken to generate the continuous data from the original observations, which includes removing inconsistent measurement and outliers, filling missing data with linear regressions to nearby stations. Details regarding the quality assurance are provided by Siemens (2016). The original measured data are also provided for these sites.

## 4 Snow survey data

### 4.1 Historical snow survey data

Snow survey data collected by CFS from seven snow courses (SC): 1, 3, 6, 8, 11, 14, and 19 are provided for the waters years from 1963 to 1986. The location of these snow courses is shown in Fig. 1, and a brief description for each snow course is listed in Table 5. Regular measurements were carried out monthly from February to June, and each course consisted of 10 staked points where snow depth and snow water equivalent were measured. In some years, measurements were conducted more than once per month, which provided more details of seasonal snow accumulation. Both monthly snow survey data from 1963 to 1986 and detailed survey data from 1963 to 1980 are included for the historical period.

### 4.2 Recent snow survey data

Snow survey data collected from transects near the recent meteorological stations: Hay Meadow, Level Forest, Upper Clearing, Upper Forest, Vista View, and Fisera Ridge are provided for nine WY from 2007 to 2016, except for the Hay Meadow in WY 2007 when no measurements were taken. The snow survey data includes snow depth, density and snow water equivalent (SWE). The snow surveys usually occur monthly during the winter accumulation period and bi-weekly to weekly during the spring melt period. Snow depth was measured by a 1-m ruler for shallow snowpack or a 3-m measuring probe for deep snowpacks, and snow density was measured using an ESC30 snow tube for shallow snowpacks or a Mount Rose snow sampler for deeper snowpacks. At each transect, snow depth was observed at 5-m intervals, and one snow density was collected for



every five depth measurements. An example of mean transect SWE from historical and recent snow surveys for alpine and montane forest sites is shown in Fig. 10.

## 5 Streamflow data

### 5.1 Historical streamflow data

Daily average streamflow ($m^3 s^{-1}$) was estimated for Cabin Creek, Middle Creek, Twin Creek, and Upper Marmot Creek for the historical period from 1963 to 1986. Streamflow measurements were made by ECCC's Water Survey of Canada at the outlets of the respective sub-basins: Cabin Creek gauge (CCG, 05BF019), Middle Creek gauge (MCG, 05BF017), Twin Creek gauge (TCG, 05BF018), and Upper Marmot Creek gauge (UMCG, 05BF020) shown in Fig. 1. Year round streamflow discharge was estimated using stage records from flow through V-notch weirs on Middle and Twin Creeks and an H-flume on

Cabin and Upper Marmot Creeks (Canadian Forestry Service, 1976; Harder et al., 2015). The Upper Marmot gauge is located higher up the Middle Creek sub-basin and captures the streamflow generated from a predominantly alpine area. The record for Upper Marmot Creek is sporadic due to the ephemeral nature of Middle Creek at this location and locations access challenges.

For the Marmot Creek outlet, streamflow was measured by ECCC at Marmot Creek basin outlet V-notch gauging station (05BF016). The streamflow data span from 1962 to 19 June 2013 and are continuous until 1986 and seasonal

thereafter. However, the gauging station was severely damaged in the June 2013 flood (Pomeroy et al., 2016), after which no measurements have been made by ECCC. The University of Saskatchewan restored discharge measurements at this site on June 26 2013 as described in the next section. The daily average streamflow data for all sub-basins and Marmot Creek can be searched and then accessed from the ECCC Water Survey of Canada "historical hydrometric data search" website at https://wateroffice.ec.gc.ca/search/historical_e.html.

**5.2 Recent streamflow data**

Recently streamflow observations were made by the University of Saskatchewan starting spring 2007 at the sub-basin outlets and at the basin outlet after June 2013 flood mentioned in Sect. 5.1. Measurements at outlets of Cabin, Middle, and Twin Creeks ceased after 2012 as all three gauging stations (and 2013 data holding dataloggers) were destroyed in June 2013. The sites are now difficult to access as the road was destroyed, the channels are unstable and access trails are covered with fallen

trees. Flow depth was continuously measured at 15-minute interval with automated pressure transducers, and velocity was manually measured with a handheld SonTek FlowTracker acoustic Doppler velocimeter every few weeks from spring to autumn. Discharge at 15-minute interval is calculated based on rating curves from continuous flow depth and manually measured velocity. Hourly average streamflow ($m^3 s^{-1}$) is estimated from the 15-minute discharge and is provided for Cabin, Middle, and Twin Creeks from 2007 to 2012, Upper Marmot Creek from 2007 to 2016 and Marmot Creek from 26 June 2013

to 2016.



## 6 Groundwater data

Three groundwater wells (GW), 301, 303, and 305, established in the 1960s and one GW, 386, established in 1988 are continuously monitored by AEP. The location of these groundwater wells is shown in Fig. 1, and brief information regarding these wells is provided in Table 6. Daily data for these groundwater wells can be searched and accessed from AEP's

"Groundwater Observation Well Network (GOWN)" website at http://environment.alberta.ca/apps/GOWN/. Access to the hourly groundwater well data can be requested from the Groundwater Information Centre at gwinfo@gov.ab.ca.

## 7 Example data

Data from the June 2013 flood is shown as an example of weather and streamflow observed in MCRB (Fig. 11). The flood event started on 18 June and ended on 24 June. Air temperature observed at Fisera Ridge station was as high as 8 °C during

rainfall on 19 June and dropped to 0.4 °C during snowfall on 21 June; the atmosphere became saturated on 18 June and stayed saturated through 21 June (Fig. 11a). Variable wind speeds were observed at the Fisera Ridge station, changing from relatively calm conditions on 18 June to 4 m s$^{-1}$ on 20 June then dropping to an average of 2 m s$^{-1}$ before peaking at 5.5 m s$^{-1}$ on 21 June (Fig. 11b). Overcast skies persisted during much of the flood event and incoming solar radiation observed at Fisera Ridge station dropped from a peak of 533 W m$^{-2}$ on 18 June to below 266 W m$^{-2}$ throughout the event and then rose to a peak of 1038

W m$^{-2}$ on 22 June (Fig. 11b). Similar depths of precipitation fell at all elevations (1436 to 2325 m) in MCRB, with about 257 mm during 19-25 June; however, this measurement was compromised as the Geonor precipitation gauge overtopped on 21 June and could not be immediately accessed for maintenance due to damaged trails and roads. During the snowfall of 21-22 June, the depth of fresh snowpack on the ground was used to estimate precipitation based on assumption of a fresh snow density of 100 kg m$^{-3}$ (Pomeroy et al. 2016). Approximately 237 mm of rainfall was measured at Fisera Ridge station during

19-25 June, and an 8-cm deep snowpack developed at Fisera Ridge on 21 June and melted after 22 June (Fig. 11c). Rainfall and snowfall rates during the event remained less than 12 mm h$^{-1}$ and were higher than 6 mm h$^{-1}$ only on 19 and 20 June, with cumulative daily totals increasing from 41 mm on 19 June to 113 mm on 20 June, and then dropping to 77 and 18 mm on 21 and 22 June, respectively. The streamflow discharge observed at outlet of Upper Marmot Creek remained below 0.6 mm h$^{-1}$ at start of the flood event on 19 June and increased steadily on 20 June, reaching a peak of 2.84 mm h$^{-1}$ at 1:00 on 21 June and

then falling to below 1 mm h$^{-1}$ after 21 June for the remaining of the flood event (Fig. 11d). Total discharge generated at the outlet of Upper Marmot Creek was estimated to be 106 mm during 19-25 June, much of which was the result of rain-on-snow in the alpine and treeline elevations.

## 8 Data availability and structure

All data presented in this paper are publically available at the Federated Research Data Repository

(https://dx.doi.org/10.20383/101.09). Headers in most data files are self-explanatory, and all data are measured in Central

Standard Time (CST) that is 6 hours behind Greenwich Mean Time (GMT-6). Meteorological data are time-series in comma delimited .txt files organized by station. Snow survey data are stored in the .xlsx files. Historical snow survey data are summarized in a single time series file. Recent snow survey data are organized by site for a water year. Recent streamflow data are time-series and are stored in .csv files and are organized by the gauge station. Additional readme files are provided for notes on missing data, data measurement periods and units, and no measurement due to wildlife interruption. Additional GIS shapefiles are provided to show locations of historical and recent hydrometeorological and hydrometric stations as well as historical and recent snow survey transects.

## 9 Relevant graduate student theses

A number of recent theses contain detailed contemporary site information for Marmot Creek Research Basin and provide results for the recent research conducted in the basin. These theses can help familiarize researchers with the basin and better understand its hydrology. Table 7 lists the theses that can be searched and accessed from University of Saskatchewan's "eCommons" website at https://ecommons.usask.ca/.

## 10 Compilation of Marmot Creek Memories, Real-time Data and Publications

The Centre for Hydrology held a 50[th] Anniversary Workshop for MCRB in February 2013 where many of the original and recent researchers gave presentations on a half-century of scientific research in the basin. The Centre has also compiled 120 MCRB publications, and provides real-time observations from many of the current meteorological stations. The workshop presentations, publications and data can be accessed here http://www.usask.ca/hydrology/MarmotBasin.php.

## 11 Summary

Data presented in this paper provide support to ongoing research in MCRB, a mountain basin located in the Front Range of Canadian Rockies. The data include 11 water years of hourly gap-filled air temperature, relatively humidity, wind speed, precipitation, incoming solar radiation, and soil temperature from 1 October 2005 to 30 September 2016 as well as 18 water years of hourly air temperature, relatively humidity, and wind speed as well as daily precipitation from 1 October 1969 to 30 September 1987. These meteorological datasets are useful for driving hydrological models and carrying out diagnostic change detection analysis in the basin. In addition, 15-minute quality controlled data including other hydrometeorological variables such as snow depth, soil temperature, and soil moisture are presented from 1 October 2005 to 30 September 2016; these data have gaps but are useful for diagnosing model performance in snow accumulation, soil moisture and temperature. Snow survey data are included for the historical period from 1963 to 1986 and the current period from 2007 to 2016. Hourly streamflow is provided for Cabin, Middle, and Twin Creeks from 2007 to 2012, Upper Marmot Creek from 2007 to 2016, and Marmot Creek



after June 2013 flood from 26 June 2013 to 2016. Daily streamflow for Cabin Creek, Middle Creek, Twin Creek, and Upper Marmot Creek from 1963 to 1986 and Marmot Creek daily streamflow from 1962 to 19 June 2013 can be obtained from the ECCC Water Survey of Canada's "historical hydrometric data search" website. In addition, data from several groundwater wells in Marmot Creek can be accessed from AEP's "Groundwater Observation Well Network (GOWN)" website. In all,

5 these long-term meteorological and hydrometric data sets are a legacy of previous and current research activities conducted in MCRB and support ongoing efforts to detect and diagnose climate change in the basin and region, examine extreme hydrometeorological events (i.e. drought and flood), and diagnosing the basin response to land cover changes caused by stressors such as insect infestations, fire and forest harvesting. This dataset ultimately serves to advance our knowledge of hydrology of the Canadian Rockies.

**Competing interests.** The authors declare that there are no conflict of interest.

## 12 Acknowledgements

The authors would like to gratefully acknowledge the funding assistance provided from the Alberta Government departments of Environment and Parks, and Agriculture and Forestry, the IP3 Cold Regions Hydrology Network of the Canadian

Foundation for Climate and Atmospheric Sciences, the Natural Sciences and Engineering Research Council of Canada through Discovery Grants, Research Tools and Instrument Grants, Alexander Graham Bell Scholarships, and the Changing Cold Regions Network, the Global Institute for Water Security, Global Water Futures and the Canada Research Chairs programme. Logistical assistance was received from the Biogeoscience Institute, University of Calgary and the Nakiska Ski Area. Field work by many graduate students in and collaborators with the Centre for Hydrology and research officers Michael Solohub,

May Guan, Angus Duncan and Greg Galloway was essential in accurate data collection in adverse conditions. Natural Resources Canada, Canadian Forest Service are the owners of the copyright of the historical meteorological and snow survey data. This paper is dedicated to the hundreds of researchers who have contributed to data collection in Marmot Creek over the last 55 years.

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



**Figure 1:** Location and contour map of the Marmot Creek Research Basin (MCRB), showing hydrometeorological stations, hydrometric stations, groundwater wells and snow courses, and ecozones of the MCRB: alpine, treeline, upper forest, forest clearing blocks, forest circular clearings, and lower forest. Note that the size and areas of circular clearings in Twin Creek are not to scale.



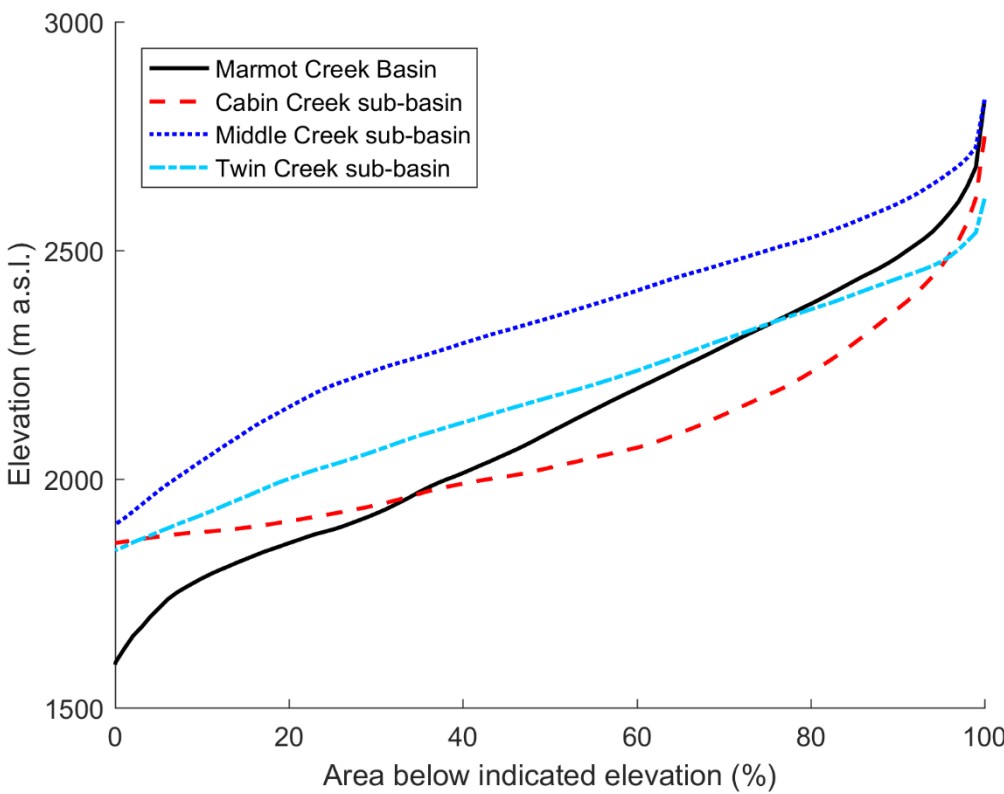

**Figure 2:** Hypsometric curves for the Marmot Creek Research Basin and three sub-basins showing the relationship between the elevation and percent area below the indicated elevation.



**Figure 3.**



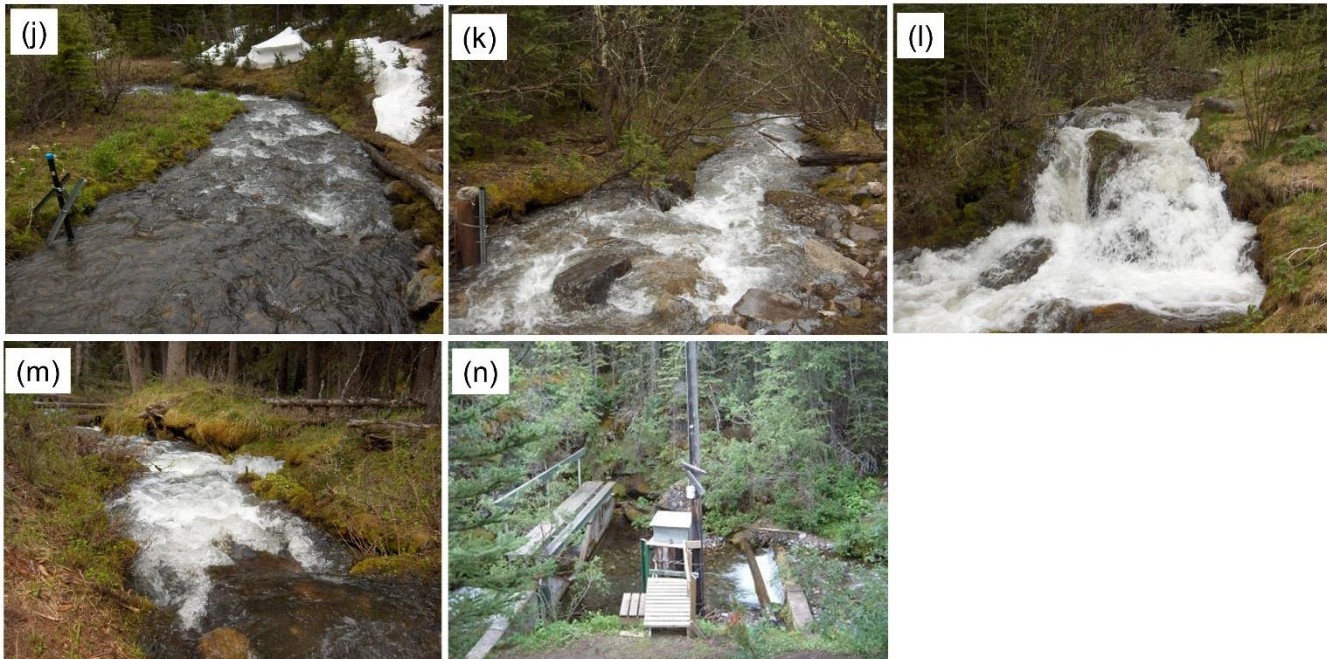

**Figure 3:** Photos of Marmot Creek Research Basin hydrometeorological and hydrometric stations: (a) Centennial Ridge in July 2010 (2470 m), (b) Fisera Ridge tripod station in April 2015 (2325 m), (c) Fisera Ridge Geonor gauge in March 2011 (2325 m), (d) Vista View in February 2011 (1956 m), (e) Upper Clearing tripod station in May 2010 (1845 m), (f) Upper Clearing Tower station in February 2011 (1845 m), (g) Upper Forest in April 2013 (1848 m), (h) Level Forest in January 2010 (1492 m), (i) Hay Meadow in February 2012 (1436 m), (j) Upper Marmot Creek stream gauge in July 2010 (2200 m), (k) Cabin Creek stream gauge in June 2010 (1710 m), (l) Middle Creek stream gauge in June 2010 (1754 m), (m) Twin Creek stream gauge in June 2010 (1754 m), (n) Marmot Creek stream gauge in June 2010 (1592 m),





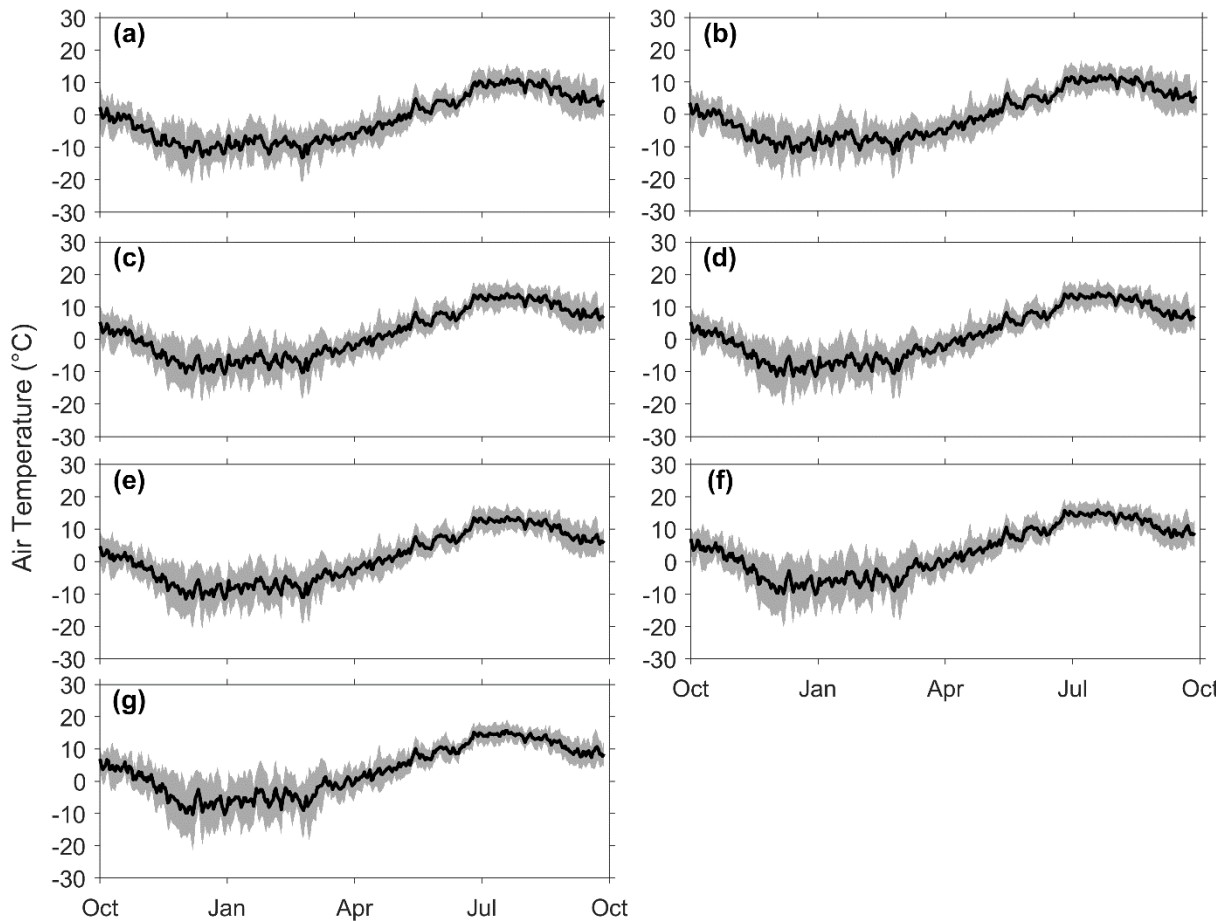

**Figure 4:** Annual mean daily air temperature for 11 water years from 1 October 2005 to 30 September 2016 at MCRB stations: (a) Centennial Ridge, (b) Fisera Ridge, (c) Vista View, (d) Upper Clearing, (e) Upper Forest, (f) Level Forest, and (g) Hay Meadow. Line represents the annual mean and the shaded area represents the standard deviation of the 11-year daily air temperature.




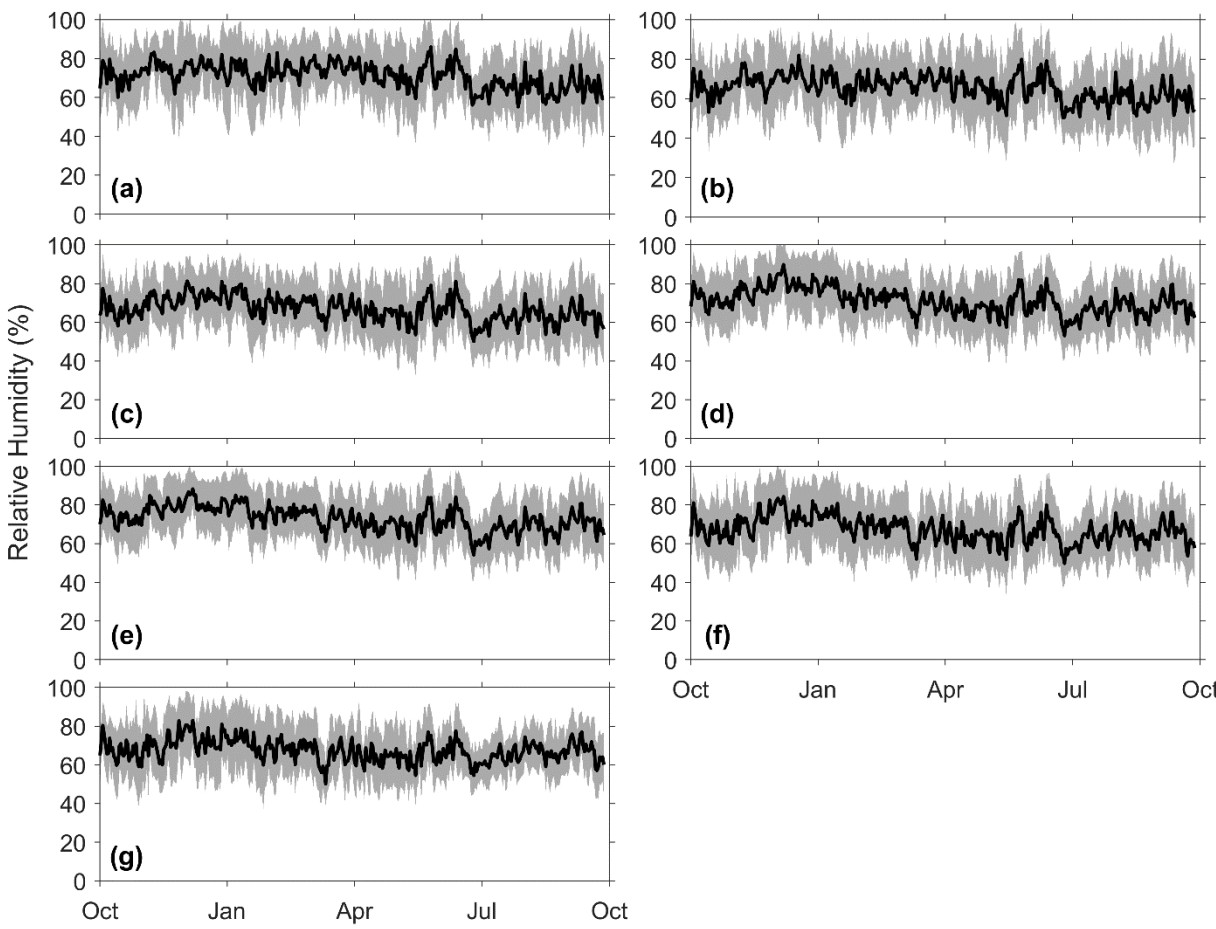

**Figure 5:** Annual mean daily relative humidity for 11 water years from 1 October 2005 to 30 September 2016 at MCRB stations: (a) Centennial Ridge, (b) Fisera Ridge, (c) Vista View, (d) Upper Clearing, (e) Upper Forest, (f) Level Forest, and (g) Hay Meadow. Line represents the annual mean and the shaded area represents the standard deviation of the 11-year daily relative humidity.





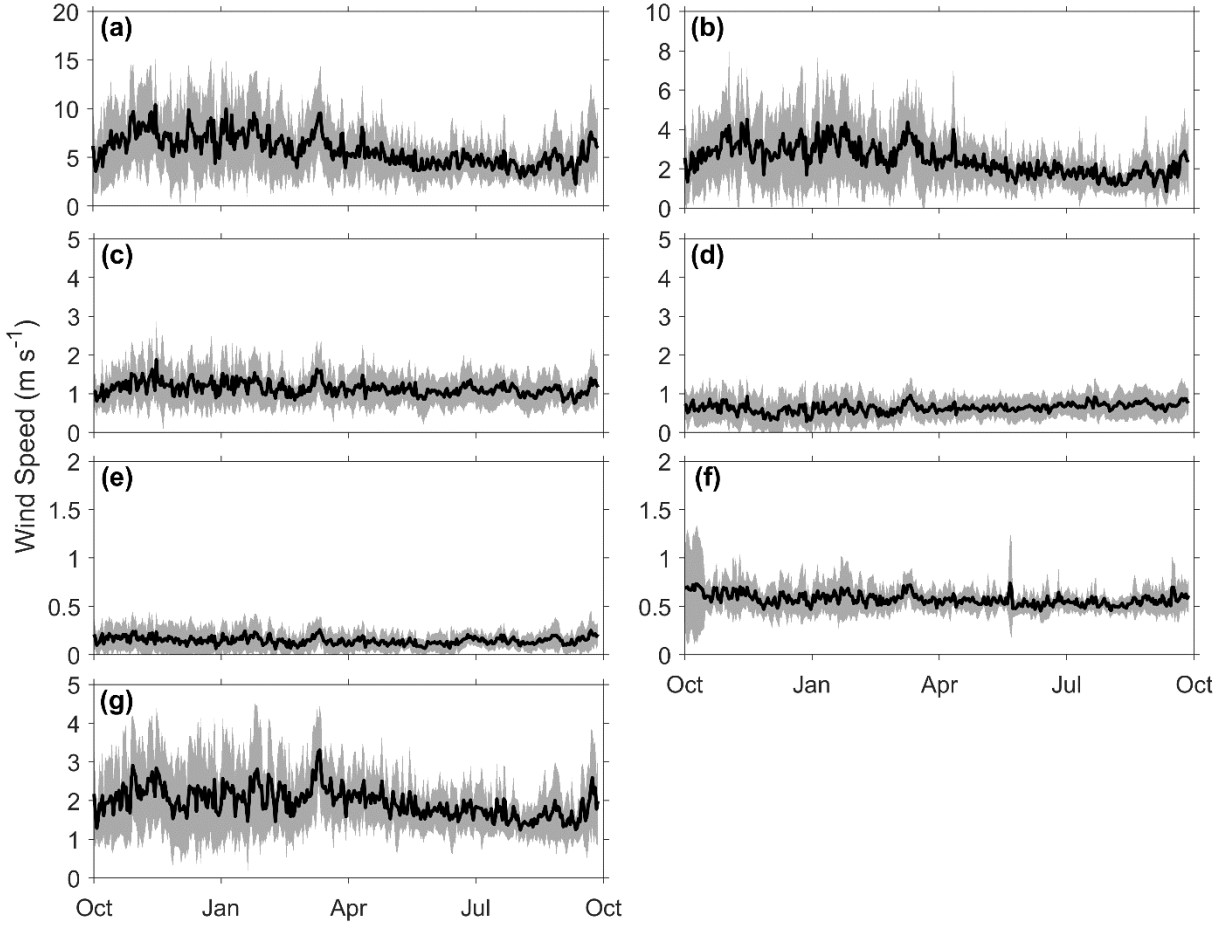

**Figure 6:** Annual mean daily wind speed for 11 water years from 1 October 2005 to 30 September 2016 at MCRB stations: (a) Centennial Ridge, (b) Fisera Ridge, (c) Vista View, (d) Upper Clearing, (e) Upper Forest, (f) Level Forest, and (g) Hay Meadow. Line represents the annual mean and the shaded area represents the standard deviation of the 11-year daily wind speed.





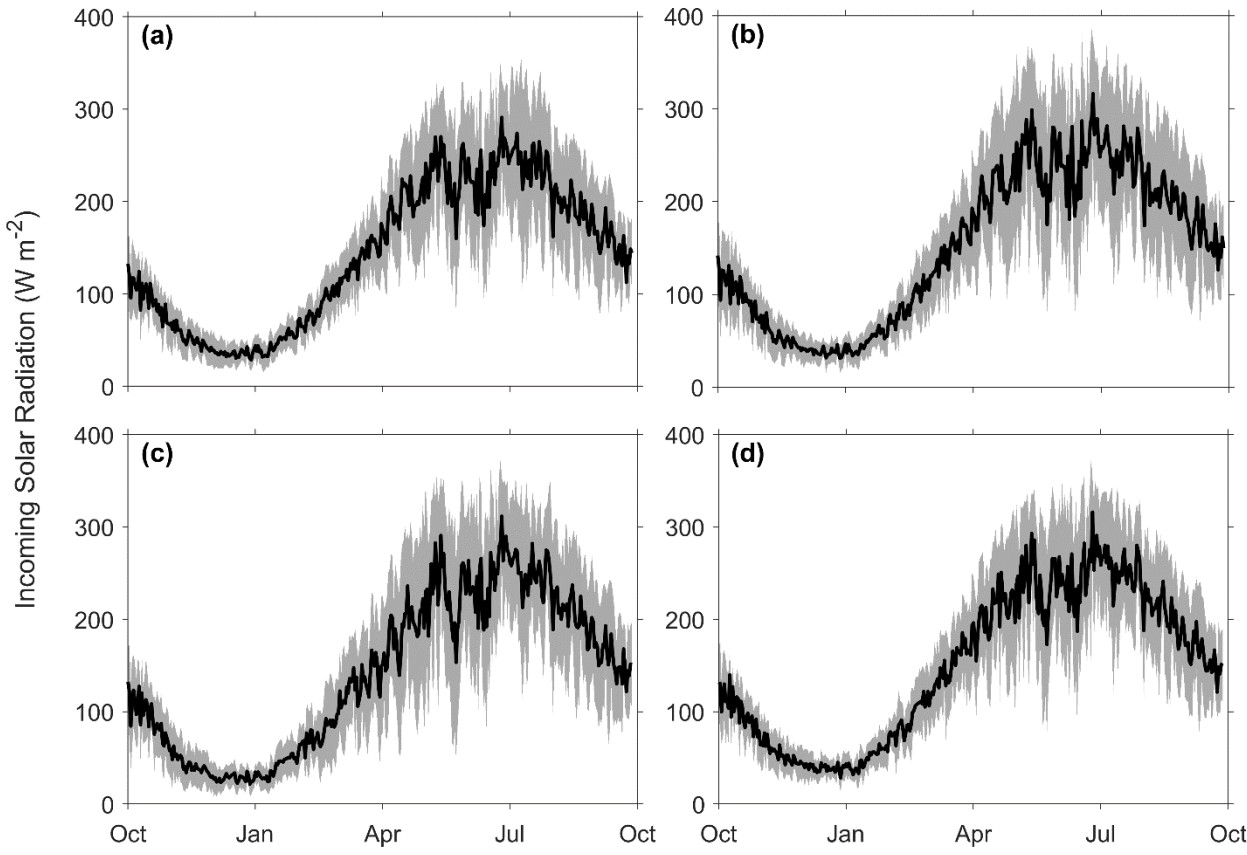

**Figure 7:** Annual mean daily incoming solar radiation for 11 water years from 1 October 2005 to 30 September 2016 at MCRB stations: (a) Centennial Ridge, (b) Fisera Ridge, (c) Upper Clearing Tower, and (d) Hay Meadow. Line represents the annual mean and the shaded area represents the standard deviation of the 11-year daily incoming solar radiation.




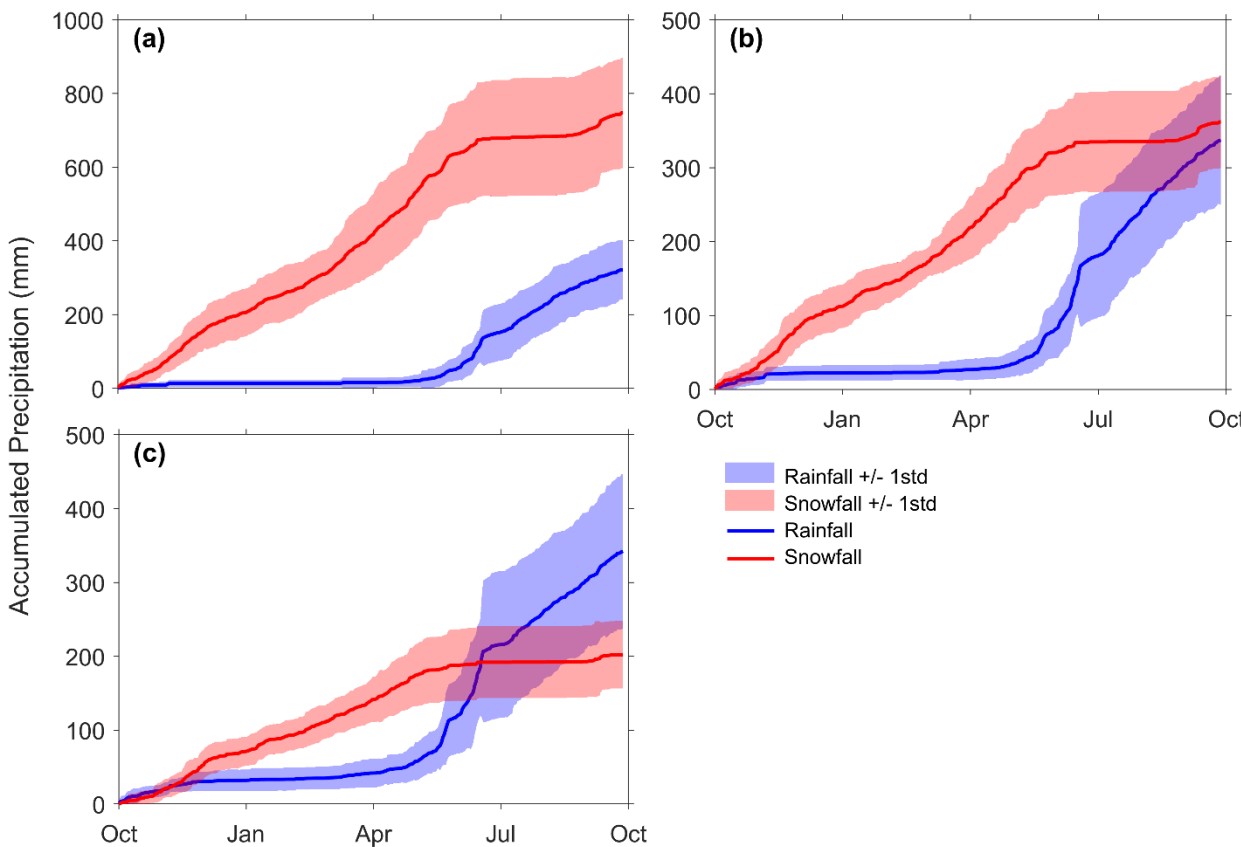

**Figure 8:** Annual mean daily accumulated rainfall and snowfall for 11 water years from 1 October 2005 to 30 September 2016 at MCRB stations: (a) Fisera Ridge, (b) Upper Clearing, and (c) Hay Meadow. Line represents the annual mean and the shaded area represents the standard deviation of the 11-year daily accumulated rainfall and snowfall. Rainfall and snowfall are calculated from wind-corrected storage-gauge observations with precipitation phase calculated as per Harder and Pomeroy (2013).





**Figure 9:** Example of hourly-averaged forcing data from Fisera Ridge station showing (a) air temperature, (b) relative humidity, (c) wind speed, (d) soil temperature, and (e) rainfall and snowfall for water years starting 1 October. All data are developed from observations except rainfall and snowfall, which are calculated from wind-corrected storage-gauge observations with precipitation phase calculated as per Harder and Pomeroy (2013).





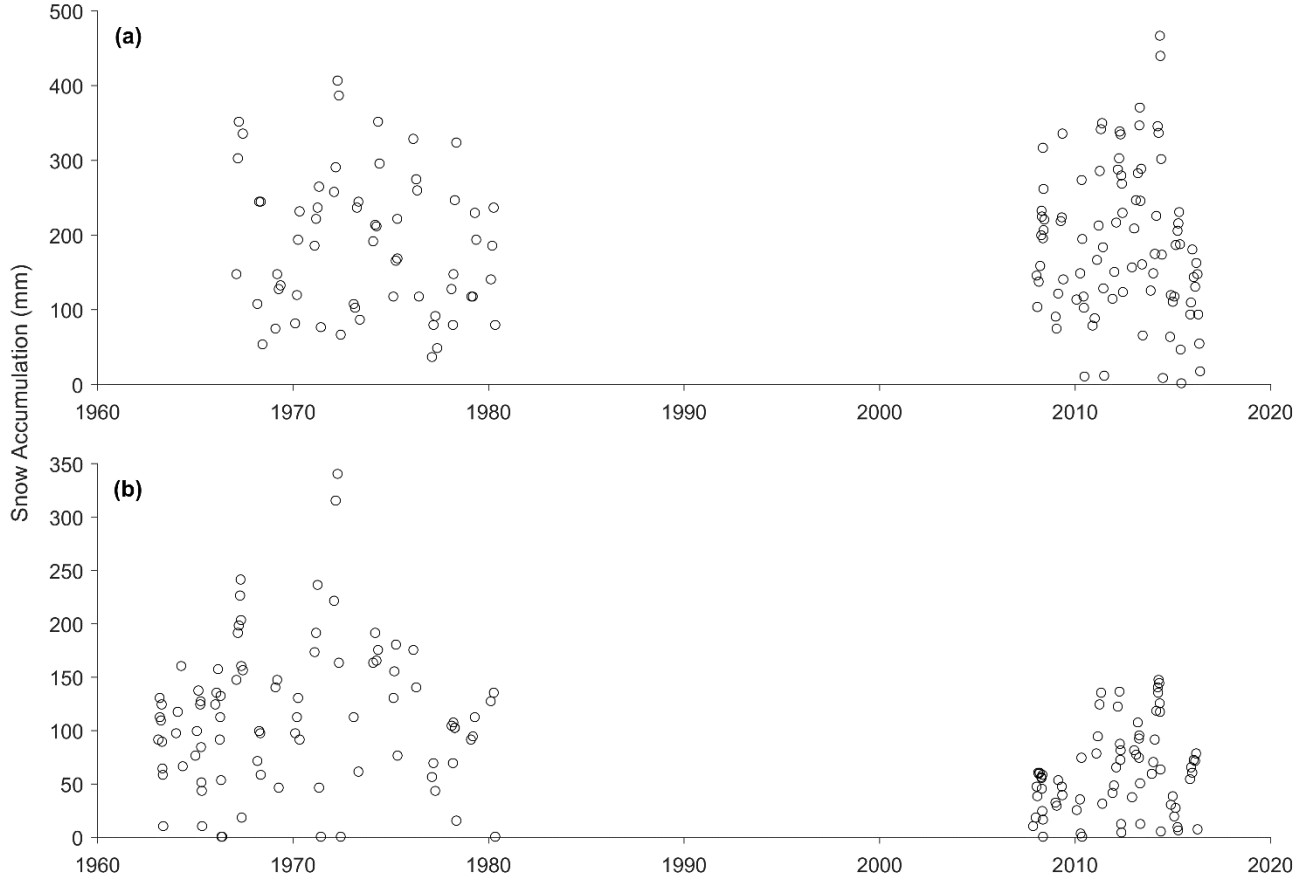

**Figure 10:** Example of mean transect snow accumulation (SWE) from (a) alpine and (b) montane forest snow survey transects. The historical SWE for alpine and montane forest is from SC 19 and SC 3 transects, respectively. The recent SWE for alpine and montane forest is from Fisera Ridge above treeline transects and Upper Clearing forest section transects, respectively.

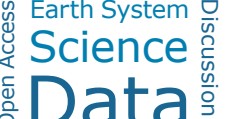



**Figure 11:** Example of hourly-averaged observations during 13-25 June 2013 from Fisera Ridge station at the Marmot Creek Research Basin showing (a) air temperature and relative humidity, (b) wind speed and incoming solar radiation, (c) rainfall and snow depth, and (d) stream discharge from Upper Marmot Creek.



**Table 1:** Area and mean elevation, aspect, and slope for ecozones at the Marmot Creek Research Basin. Note that the aspect is in degree clockwise from North.

| Ecozone | Area (km²) | Area Fraction (% of basin) | Elevation (m. a.s.l.) | Aspect (°) | Slope (°) |
|---|---|---|---|---|---|
| Alpine | 3.23 | 34.5 | 2413 | 110 | 30 |
| Treeline | 0.93 | 10.0 | 2217 | 91 | 22 |
| Upper Forest | 2.75 | 29.3 | 1983 | 108 | 20 |
| Forest Clearing Blocks | 0.40 | 4.3 | 1927 | 140 | 11 |
| Forest Circular Clearing North-facing | 0.26 | 2.7 | 1966 | 34 | 17 |
| Forest Circular Clearing South-facing | 0.24 | 2.6 | 2014 | 113 | 21 |
| Lower Forest | 1.42 | 15.2 | 1756 | 113 | 14 |



**Table 2:** Hydrometeorological variables, instrumentation and height from the recent stations at the Marmot Creek Research Basin. AGS and BGS denote the distance above ground surface and below ground surface, respectively; n/a denotes not applicable.

| Station | Hay Meadow | Level Forest | Upper Clearing | Upper Clearing Tower | Upper Forest | Vista View | Fisera Ridge | Centennial Ridge |
|---|---|---|---|---|---|---|---|---|
| Coordinates | 50.9441°N; 115.1389°W, 1436 m | 50.9466°N; 115.1464°W, 1492 m | 50.9565°N; 115.1754°W, 1845 m | 50.9565°N; 115.1754°W, 1845 m | 50.9569°N; 115.1762°W, 1848 m | 50.9712°N; 115.1722°W, 1956 m | 50.9560°N; 115.2041°W, 2325 m | 50.9571°N; 115.1930°W, 2470 m |
| Record | 1 October 2005-30 September 2016 | 10 March 2005-30 September 2016 | 7 June 2005-30 September 2016 | 21 October 2007-30 September 2016 | 7 June 2005-30 September 2016 | 1 September 2005-30 September 2016 | 13 October 2006-30 September 2016 | 24 July 2005-30 September 2016 |
| Air Temperature (°C) and Relative Humidity (%) | Vaisala HMP45C212 | Vaisala HMP45C212 | Vaisala HC2-S3 | Vaisala HMP45C212 | Vaisala HMP45C212 | Vaisala HMP45C212 | Vaisala HMP45C212 | Vaisala HMP45C212 |
| AGS (m) | 1.86 | 2.27 | 2.15 | 17 | 2.33 | 2.74 | 2.3 | 1.93 |
| Wind Speed (m s⁻¹) and Wind Direction (degree) | RM Young 05305-10 Wind Monitor | Met One 50.5 Sonic Anemometer | RM Young 05305-10 Wind Monitor | RM Young 05305-10 Wind Monitor | RM Young 05305-10 Wind Monitor | RM Young 05105-10 Wind Monitor | Wind Speed and Direction A - RM Young 05305-10 Wind Monitor Wind Speed B - 3-cup anemometer | RM Young 05105-10 Wind Monitor |
| AGS (m) | 7 | 2.45 | 2.85 | 18 | 2.77 | 4.11 | A - 2.55 B - 4.2 | 2.41 |
| Snow Depth (m) | SR50 | SR50 | SR50 | n/a | SR50 | SR50 | SR50 | SR50 |
| AGS (m) | 1.65 | 1.04 | 1.76 | | 1.63 | 1.59 | 1.19 | 1.03 |
| Soil Temperature (°C) | K-type Thermocouple | K-type Thermocouple | K-type Thermocouple | n/a | K-type Thermocouple | K-type Thermocouple | CS 107B Thermistor | CS 107B Thermistor |
| BGS (cm) | A - 5 | A - 5 | A - 10 | | A - 10 | A - 5 | A - 5 | A - 5 |
| | B - 10 | B - 25 | B - 20 | | B - 20 | B - 10 | B - 15 | B - 15 |
| | C - 20 | C - 40 | | | | C - 20 | | |
| Soil Heat Flux (W m⁻²) | HFT3 Heatflux Plate | HFT3 Heatflux Plate | HFP01 Heatflux Plate | n/a | n/a | HFP01 Heatflux Plate | HFT3 Heatflux Plate | n/a |
| BGS (cm) | 10 | 10 | 10 | | | 2 | 10 | |
| Soil Moisture (m³ m⁻³) | CS616 Soil Moisture Probe | CS616 Soil Moisture Probe | n/a | n/a | CS616 Soil Moisture Probe | n/a | n/a | n/a |
| BGS (cm) | 15 | 25 | | | 25 | | | |
| Incoming Solar Radiation (W m⁻²) Outgoing Solar Radiation (W m⁻²) | Kipp and Zonen CM3 Pyranometers | Kipp and Zonen CM3 Pyranometers | Kipp and Zonen CM3 Pyranometers | Kipp and Zonen CM21 Pyranometer, 20 | Kipp and Zonen CM3 Pyranometers | Apogee CS300-L Pyranometer, 1.97 | Kipp and Zonen CM3 Pyranometers | Licor LI200s Shortwave Radiometer |
| AGS (m) | 1.95 | 1.31 | 2.33 | n/a | 1.95 | n/a | 1.45 | 1.37 |
| Incoming Longwave Radiation (W m⁻²) Outgoing Longwave Radiation (W m⁻²) | Kipp and Zonen CG3 Pyrgeometers | Kipp and Zonen CG3 Pyrgeometers | Kipp and Zonen CG3 Pyrgeometers | Kipp and Zonen CG1 Pyrgeometer, 20 | Kipp and Zonen CG3 Pyrgeometers | n/a | Kipp and Zonen CG3 Pyrgeometers | n/a |
| AGS (m) | 1.95 | 1.31 | 2.33 | n/a | 1.95 | n/a | 1.45 | n/a |
| Rainfall (mm) | Texas TE525M Rain Gauge | n/a | Hydrological Services TB4 Tipping Bucket Rain Gauge | n/a | Texas TE525M Rain Gauge | n/a | Hydrological Services TB4 Tipping Bucket Rain Gauge | Texas TE525M Rain Gauge |
| AGS (m) | 2.56 | | 2.36 | | 0.7 | | 4.2 | 1.56 |
| All Precipitation (mm) | Geonor T200B Gauge with Alter Shield | n/a | Geonor T200B Gauge with Alter Shield | n/a | n/a | n/a | Geonor T200B Gauge with Alter Shield | n/a |
| AGS (m) | 1.8 | | 1.85 | | | | 4.1 | |
| Barometric Pressure (mb) | BP61025V Pressure Sensor | n/a | CS106 Barometric Pressure Sensor | n/a | n/a | n/a | n/a | BP61025V Pressure Sensor |
| AGS (m) | 1.25 | | 1.25 | | | | | 0.7 |



**Table 3:** Quality controlled threshold values for 15-minute hydrometeorological variables for current stations in MCRB; ROC and n/a denote rate of change and not applicable, respectively.

| Variable | Unit | Maximum | Minimum | ROC limit | Time steps to flag constant value |
|---|---|---|---|---|---|
| Air Temperature | °C | 40 | -60 | 10 | 16 |
| Relative Humidity | % | 100 | 0 | 30% | 16 |
| Wind speed | m s$^{-1}$ | 30 | 0 | n/a | 16 |
| Snow Depth | m | 5 | 0 | n/a | n/a |
| Soil Temperature | °C | 50 | -40 | 10 | 96 |
| Soil Heat Flux | W m$^{-2}$ | 1000 | -500 | 100 | 16 |
| Soil Moisture | fraction | 1 | 0 | 0.2 | 16 |
| Solar Radiation | W m$^{-2}$ | 1368 | 0 | 1450 | 48 |
| Longwave Radiation | W m$^{-2}$ | 600 | 100 | 300 | 16 |
| Precipitation | mm | 30 | 0 | n/a | n/a |
| Barometric pressure | mb | 1090 | 650 | 30 | 16 |



**Table 4:** Mean water year air temperature and total water year precipitation from the current stations at the Marmot Creek Research Basin. Values inside parentheses are total water year snowfall.

| Water Year | Mean Air Temperature (°C) | | | | | | | Total Precipitation (mm) | | |
| | Centennial Ridge | Fisera Ridge | Vista View | Upper Clearing | Upper Forest | Hay Meadow | Level Forest | Fisera Ridge | Upper Clearing | Hay Meadow |
|---|---|---|---|---|---|---|---|---|---|---|
| 2006 | -1.2 | 0.1 | 2.9 | 2.3 | 1.7 | 4.0 | 4.0 | 902 (551) | 646 (306) | 492 (155) |
| 2007 | -1.7 | -0.5 | 2.2 | 1.5 | 0.8 | 3.3 | 3.4 | 1215 (815) | 797 (421) | 631 (196) |
| 2008 | -2.7 | -1.7 | 0.8 | 0.6 | 0.0 | 2.3 | 2.4 | 1218 (926) | 804 (421) | 693 (231) |
| 2009 | -1.4 | -0.7 | 1.4 | 1.0 | 0.4 | 2.8 | 2.8 | 944 (638) | 610 (332) | 450 (210) |
| 2010 | -2.1 | -1.0 | 0.6 | 0.6 | 0.0 | 2.6 | 2.4 | 1140 (904) | 670 (410) | 476 (205) |
| 2011 | -2.4 | -1.2 | 0.4 | 0.4 | -0.2 | 1.9 | 1.8 | 1128 (865) | 671 (396) | 522 (271) |
| 2012 | -1.4 | -0.2 | 1.5 | 1.6 | 1.0 | 3.6 | 3.6 | 1247 (922) | 794 (419) | 586 (201) |
| 2013 | -1.5 | -0.2 | 1.3 | 1.4 | 1.1 | 3.1 | 3.0 | 1329 (794) | 868 (320) | 762 (207) |
| 2014 | -2.1 | -0.8 | 0.7 | 0.5 | 0.2 | 2.2 | 2.1 | 877 (658) | 650 (419) | 510 (267) |
| 2015 | -0.4 | 0.9 | 2.3 | 2.5 | 2.2 | 4.2 | 4.2 | 857 (543) | 593 (272) | 440 (163) |
| 2016 | -0.3 | 1.0 | 2.4 | 2.7 | 2.4 | 4.4 | 4.5 | 939 (614) | 591 (268) | 426 (118) |
| 11-water year mean | -1.6 | -0.4 | 1.5 | 1.4 | 0.9 | 3.1 | 3.1 | 1070 (748) | 699 (362) | 545 (202) |



**Table 5:** Historical snow courses (SC) at the Marmot Creek Research Basin from description by Fisera (1977).

| Snow Course | Description |
|---|---|
| 1 | East sloping lodgepole pine about 9m tall with natural openings |
| 3 | Gently south sloping mature spruce, lodgepole pine and alpine |
| 6 | Gently northeast sloping mature spruce, lodgepole pine and alpine fir |
| 8 | South sloping lodgepole pine about 6m tall |
| 11 | Southeast sloping mature spruce, lodgepole pine and alpine fir |
| 14 | Northeast sloping mature spruce, lodgepole pine and alpine fir with small natural openings |
| 19 | Variable terrains (i.e. north and south slope, flat and gullies) above treeline |



**Table 6:** Active groundwater wells (GW) at the Marmot Creek Research Basin.

| GW Well | Station Name | Established | Elevation (m) | Depth (m) | Aquifer | Lithology |
|---|---|---|---|---|---|---|
| 301 | Marmot Creek Basin S5250_0301 | 11 October 1964 | 1601.4 | 12.2 | Rocky Mountain | Sandstone |
| 303 | Marmot Creek Basin N5475_0303 | 9 July 1965 | 1669.1 | 36.58 | Rocky Mountain | Sandstone |
| 305 | Marmot Creek Basin N6770_0305 | 14 July 1965 | 2063 | 11.58 | Fernie | Shale |
| 386 | Marmot Creek Basin N2507E_0386 | 18 November 1988 | 1894 | 12.8 | Surficial | Gravel and Clay |



**Table 7:** Marmot Creek Research Basin theses in chronological order for the recent period.

| Thesis Title | Author | Year |
|---|---|---|
| Compositional change of meltwater infiltrating frozen ground | Lilbæk, Gro | 2009 |
| Energy fluxes at the air-snow interface | Helgason, Warren | 2009 |
| Unloading of intercepted snow in conifer forests | MacDonald, James | 2010 |
| Hydrological response unit-based blowing snow modelling over mountainous terrain | MacDonald, Matthew | 2010 |
| Radiation and snowmelt dynamics in mountain forests | Ellis, Chad | 2011 |
| Simulating areal snowcover depletion and snowmelt runoff in alpine terrain | DeBeer, Chris | 2012 |
| Implications of mountain shading on calculating energy for snowmelt using unstructured triangular meshes | Marsh, Christopher | 2012 |
| Precipitation phase partitioning with a psychrometric energy balance: model development and application | Harder, Phillip | 2013 |
| Acoustic measurement of snow | Kinar, Nicholas | 2013 |
| Effects of climate variability on hydrological processes in a Canadian Rockies headwaters catchment | Siemens, Evan | 2016 |
| Sensitivity analysis of mountain hydrology to changing climate | Rasouli, Kabir | 2017 |