# Peer review of "Hydrometeorological data from Marmot Creek Research Basin, Canadian Rockies"

_Earth System Science Data, 2018_

## Referee Comment (RC1) · Anonymous Referee #1 · 19 Nov 2018

OVERVIEW

This paper presents an impressive compilation of data from the Marmot Creek Research Basin (MCRB) from two separate periods, the first being 1962 to 1986, and the second from 2005 onward. The research site has been subject to numerous studies, a review of which is provided in the introduction, which is both interesting to read and potentially helpful for authors of future studies. The data description is detailed, Table 2 provides a nice example of meta data available for the more recent instrumentation. The portal that hosts the datasets is straight forward to use, and files are easily downloaded after two, three mouse clicks.

Overall, a nice and thorough presentation. I see one important shortcoming. But all other comments and suggestions are either minor and/or a matter of taste.

[Figure]

MAJOR COMMENT

If this data is disseminated to allow users "developing hydrological process understanding, evaluating process algorithms and hydrological, cryospheric or atmospheric models", then we need more information about the catchment itself. While a DEM might be easily available to most potential users of this data, how are they supposed to inform their models, e.g., about canopy processes? The data compilation seems incomplete without detailed information about variables such as LAI and canopy closure, in particular given that MCRB was subject to forest management experiments. There is also no information whether the clearings are maintained to remain open or if they are overgrown by now.

Further, "Snow survey data [were] collected from transects near the recent meteorological stations". Given the images in Figure 3 c-f, these data could be collected inside the clearing, in the forest, or across the forest edge. But without having more detailed information it is difficult to use the snow course data for model validation purposes.

I am sure these information are available in one or several of the publications cited in the manuscript (maybe Hopkinson?). But as a user I don't want to read them all before eventually finding what I need. Similar consideration go with soil data.

I guess this shortcoming is easy to fix, but I would ask the authors to reassess their manuscript from the perspective of a modeler who is unfamiliar with the site and does not know how to access auxiliary data needed to set up a meaningful model application.

MINOR COMMENTS AND SUGGESTIONS, reference is given to [page / line number]

[2 / 4-7] split this sentence into two.

[2 / 11] refer to Figure 1.

[3/12-14] please move this sentence to the above section with the literature review. This paragraph here should describe the content of this paper only.

[Figure]

[4/23] it might be useful to mention what percentage of the data had to be removed (which seems a fairly basic descriptor of a dataset).

[4/27] modelers use different time steps for their models. So I would not necessarily call hourly data "modelling data".

[5/1] are these gap-filled data identifiable? If so by what means?

[5/9] apart from gap filling, I am not sure "estimated data" should be included in this data assembly.

[5/19] ventilated? radiation shield?

[6/9] what is "due to the length of measurement" supposed to mean?

[7/16] I would go by the same order as the previous section. Recent data first, historical then. Or the other way around, but be consistent.

[8/4] and [8/20] some info on the discharge measurements should be added. Is there a maximum capacity of the V-notch? Until what flow level is the streamflow data safe to use, the rating curve established, respectively? Was the stationarity of the rating curve monitored? Looking at Figure 3, more info is certainly needed.

[8/23] replace "after 2012" by "in June 2013".

[10/8] consider merging sections 9 and 10.

---

## Referee Comment (RC2) · Anonymous Referee #2 · 1 Feb 2019

**GENERAL COMMENTS**

This manuscript describes two long-term datasets, (1) a historical time period (1962–1987) and (2) a modern time period (2005–2016), from the Marmot Creek Research Basin in the Canadian Rockies. These data provide much-needed insight to changing weather patterns in northern high-altitude regions, and could easily be used to perform a number of important modeling and climate sensitivity studies. The authors' description of the datasets is concise and coherent, and the paper is structured in a way that makes it easy to read. I recommend this manuscript to be published once these minor revisions detailed below are addressed.

1. Since this seems to be the defining hydrometeorological dataset for the MCRB, it would be helpful if some of the spatial information necessary for hydrological analysis

[Figure]

of the basin were delivered alongside. Then any researcher looking to use these data for a spatial modeling study would not have to look elsewhere and derive their own digital elevation models, vegetation masks, basin masks, stream networks, etc.

2. The README.txt file contains a huge amount of information about the individual files. However, much of the information is repeated ad nauseum making this README file almost impossible to navigate. My suggestion would be to remove the repeated information and/or consider using markdown language to make the information easier to comb through.

3. The figures and tables include site description information, but the dataset itself has no mention of these important metadata. A brief paragraph within the README file describing where this site information is located would be very helpful.

4. The purpose of the **publication_9-2018-10-16-22-08-40-sha256-sums.txt** file within the dataset folder is not apparent.

5. There are a great deal of acronyms in this manuscript. An appendix listing the acronym definitions just before the References section would help some of the page flipping and searching.

**SPECIFIC COMMENTS**

pg. 2, line 24 - AEP is not previously defined.

pg. 3, Site description section - This section is all one paragraph but could benefit from being split into two, three, or even four individual paragraphs.

pg. 6, line 4 - I have to question these reported wind speed measurements. From the specifications of the R.M. Young 05305 anemometer, the threshold sensitivity of the instrument is 0.4 m/s. The wind speeds in the sheltered sites seem to be below the measurement threshold that the sensor can measure. In my experience, wind speed measurements should be capped at a lower limit of around 0.4 m/s due to limitations of the internal bearings that cause inherent noise in the data.
pg. 6, line 11-15 - When calculating the mean incoming solar radiation, are nighttime hours included?

pg. 7, line 21-22 - The last sentence of this paragraph is unclear. When you say 'detailed survey data', are you just referring to occasional months with two measurements? The reader could also take that to mean the data include individual hole-by-hole SWE measurements, which is not provided here.

Figures 4–8 - Include some light gridlines in these plots make them easier to comprehend.

Figure 10 - The data in each of these plots are not from the same stations, so I suggest making them different marker types and including a legend.

**TECHNICAL CORRECTIONS**

pg. 8, line 12 - Replace 'locations access challenges' with 'site access challenges'.

pg. 11, line 7 - '...diagnose the basin response...'

Table 2 - It seems that there are problems with the reported AGS of the incoming solar radiation row for the Upper Clearing Tower and Vista View columns. For instance, below the 'Kipp and Zonen CM21 Pyranometer' there is a hanging '20', which looks like it should go below where you have an 'n/a'.

---

## Author Comment (AC1) · 1 Mar 2019

Response to referees' comments on "Hydrometeorological data from Marmot Creek Research Basin, Canadian Rockies" by Xing Fang, John W. Pomeroy, Chris M. DeBeer, Phillip Harder, and Evan Siemens

Response to Anonymous Referee #1's comments:

OVERVIEW

This paper presents an impressive compilation of data from the Marmot Creek Research Basin (MCRB) from two separate periods, the first being 1962 to 1986, and the second from 2005 onward. The research site has been subject to numerous studies, a review of which is provided in the introduction, which is both interesting to read and

potentially helpful for authors of future studies. The data description is detailed, Table 2 provides a nice example of meta data available for the more recent instrumentation. The portal that hosts the datasets is straight forward to use, and files are easily downloaded after two, three mouse clicks. Overall, a nice and thorough presentation. I see one important shortcoming. But all other comments and suggestions are either minor and/or a matter of taste.

Response: Thanks reviewer #1 for the overview comment. It is encouraging to hear that when trying to assemble data collected at many stations and from different periods.

MAJOR COMMENT

If this data is disseminated to allow users "developing hydrological process understanding, evaluating process algorithms and hydrological, cryospheric or atmospheric models", then we need more information about the catchment itself. While a DEM might be easily available to most potential users of this data, how are they supposed to inform their models, e.g., about canopy processes? The data compilation seems incomplete without detailed information about variables such as LAI and canopy closure, in particular given that MCRB was subject to forest management experiments. There is also no information whether the clearings are maintained to remain open or if they are overgrown by now.

Response: To keep this paper concise, we presented sections listing relevant graduate student theses and a website storing publications (e.g reports, peer-review journal articles) for Marmot Creek Research Basin which contain many details on the basin. To fully describe the basin we have added a DEM and digital forest cover map for the basin and have mentioned this in new sentences Section 2 Site Description. We have added information in the Section 2 Site Description on the current status of the old forest clearings and clear-cut blocks, which have regrown as sparse juvenile forests to varying degrees.

Further, "Snow survey data [were] collected from transects near the recent meteorological stations". Given the images in Figure 3 c-f, these data could be collected inside the clearing, in the forest, or across the forest edge. But without having more detailed information it is difficult to use the snow course data for model validation purposes. I am sure these information are available in one or several of the publications cited in the manuscript (maybe Hopkinson?). But as a user I don't want to read them all before eventually finding what I need. Similar consideration go with soil data. I guess this shortcoming is easy to fix, but I would ask the authors to reassess their manuscript from the perspective of a modeler who is unfamiliar with the site and does not know how to access auxiliary data needed to set up a meaningful model application.

Response: In the dataset, there is a "Station_Shapefile" folder, in that we provide a "Recent_snow_survey_transects" folder including GIS shapefiles for all recent snow survey transects. With the shapefiles, users can view the transects in GIS software and find the location of transects when using snow survey for model validation purpose. In addition, the recent snow survey data contains field notes on weather condition and landcover information of each snow survey transect, and they are readily to users. We added a description in "Recent snow survey data" section to inform readers about that in addition to snow depth, density and snow water equivalent, snow survey data contains field notes on the land cover of the snow survey transect. Regarding on soil data information, a detailed soil survey dataset was published in Beke's Ph.D. thesis in 1969. The thesis has soil data information for several sites at Marmot Creek and the thesis can be downloaded from website link we provided in publication link.

MINOR COMMENTS AND SUGGESTIONS, reference is given to [page / line number]

[2 / 4-7] split this sentence into two.

Response: Yes, it is split into two sentences.

[2 / 11] refer to Figure 1.

Response: Yes, a reference to Figure 1 is added in the sentence.

[3/12-14] please move this sentence to the above section with the literature review. This paragraph here should describe the content of this paper only.

Response: This sentence shows examples of studies that have already used the dataset, and we think it is better in this paragraph than the previous paragraph that provides generally literature on recent research activities at Marmot Creek.

[4/23] it might be useful to mention what percentage of the data had to be removed (which seems a fairly basic descriptor of a dataset).

Response: In the quality control (QC) procedure, we used a script that does QC removal based on threshold values. In this script, the removed data and missing data in raw data are both flagged as -9999, and the value -9999 is used by subsequent QC scripts. It is not easy to calculate the percentage with current QC scripts. However, reviewer provided a valuable comment, and we will incorporate this into QC scripts for the future data cleanup work.

[4/27] modelers use different time steps for their models. So I would not necessarily call hourly data "modelling data".

Response: The modelling data we provided is in hourly time step. We wrote it to inform users and readers that modelling data is hourly and is the average from the 15-minute QC data, and the missing data in QC is filled for the hourly modelling data.

[5/1] are these gap-filled data identifiable? If so by what means?

Response: The gap-filled data are identifiable. We provided the hourly modelling data that is the average from the 15-minute QC data. The hourly modelling data is gap-filled, and the 15-minute QC data is the data with gap, with denoted value of -9999.

[5/9] apart from gap filling, I am not sure "estimated data" should be included in this data assembly.

Response: Yes, this is the hourly modelling data that used to forcing modelling project

at Marmot Creek. The modelling project requires data from several stations at the same time periods, thus we estimated the data for these two stations for the time periods before their establishment. We provided this write-up in this paragraph to inform users that they are "estimated" modelling data, so users can decide whether to use the data or not.

[5/19] ventilated? radiation shield?

Response: Yes, they are naturally ventilated Gill radiation shields. This is added to the sentence.

[6/9] what is "due to the length of measurement" supposed to mean?

Response: We reworded it. Hourly modelling of incoming solar radiation is not provided for Vista View station due to the short length of measurement.

[7/16] I would go by the same order as the previous section. Recent data first, historical then. Or the other way around, but be consistent.

Response: Yes, we changed the order and placed the recent snow survey data first and then historical snow survey data.

[8/4] and [8/20] some info on the discharge measurements should be added. Is there a maximum capacity of the V-notch? Until what flow level is the streamflow data safe to use, the rating curve established, respectively? Was the stationarity of the rating curve monitored? Looking at Figure 3, more info is certainly needed.

Response: this V-notch weir is described in detail as an example of a well-gauged stream in Bruce and Clark (1965) Introduction to Hydrometeorology . It was operated by the Water Survey of Canada according to national standards which include periodic updating of the rating curve. Its capacity was never exceeded by discharge until it was destroyed in the flood event of June 2013, where it was filled with debris and the channel diverted to flow beside the weir. The post June 2013 streamflow discharge data is calculated from rating curves developed by the University of Saskatchewan that

are frequently checked by manual measurements throughout the spring, summer and fall. We added a sentence to clarify this.

[8/23] replace "after 2012" by "in June 2013".

Response: Yes, we replaced it as suggested.

[10/8] consider merging sections 9 and 10.

Response: Yes, we merged these two sections.

Response to Anonymous Referee #2's comments:

GENERAL COMMENTS

This manuscript describes two long-term datasets, (1) a historical time period (1962–1987) and (2) a modern time period (2005–2016), from the Marmot Creek Research Basin in the Canadian Rockies. These data provide much-needed insight to changing weather patterns in northern high-altitude regions, and could easily be used to perform a number of important modeling and climate sensitivity studies. The authors' description of the datasets is concise and coherent, and the paper is structured in a way that makes it easy to read. I recommend this manuscript to be published once these minor revisions detailed below are addressed. 1. Since this seems to be the defining hydrometeorological dataset for the MCRB, it would be helpful if some of the spatial information necessary for hydrological analysis of the basin were delivered alongside. Then any researcher looking to use these data for a spatial modeling study would not have to look elsewhere and derive their own digital elevation models, vegetation masks, basin masks, stream networks, etc.

Response: We have added spatial information for MCRB – specifically a DEM, vegetation masks and basin masks and stream networks derived from the DEM to a Basin_GIS folder on the FRDR site.

2. The README.txt file contains a huge amount of information about the individual

files. However, much of the information is repeated ad nauseum making this README file almost impossible to navigate. My suggestion would be to remove the repeated information and/or consider using markdown language to make the information easier to comb through.

Response: We publish and deposit the dataset to the Federated Research Data Repository (FRDR). The README file serves as a metadata and the huge amount of information about individual files is required by FRDR according to its regulations and protocols. The information about individual files seems repetitive, but it is used to provide metadata for each file under different folder, and it is organized in the way sanctioned by FRDR.

3. The figures and tables include site description information, but the dataset itself has no mention of these important metadata. A brief paragraph within the README file describing where this site information is located would be very helpful.

Response: The figure and table used for site descriptions are to provide readers information on station location, land cover and elevation of basin when readers plan to use data. The README file is used to document metadata information for the data we publish on Federated Research Data Repository according to its regulation and protocol. The paper supplements the README file, but the README file is not to describe the paper. The information used for the site description figure and table was derived from many recent publications which detail this. And it is available in the basin DEM and vegetation cover maps which are now published with this paper.

4. The purpose of the publication_9-2018-10-16-22-08-40-sha256-sums.txt file within the dataset folder is not apparent.

Response: We do not own this file. This file is automatically generated by the Federated Research Data Repository (FRDR) when the dataset is published. It is a record file from FRDR.

5. There are a great deal of acronyms in this manuscript. An appendix listing the acronym definitions just before the References section would help some of the page flipping and searching.

Response: Yes, we added an appendix show a list of acronym before the References section.

SPECIFIC COMMENTS

pg. 2, line 24 - AEP is not previously defined.

Response: Yes, we added the definition for AEP.

pg. 3, Site description section - This section is all one paragraph but could benefit from being split into two, three, or even four individual paragraphs.

Response: Yes, we split the site description into four paragraphs.

pg. 6, line 4 - I have to question these reported wind speed measurements. From the specifications of the R.M. Young 05305 anemometer, the threshold sensitivity of the instrument is 0.4 m/s. The wind speeds in the sheltered sites seem to be below the measurement threshold that the sensor can measure. In my experience, wind speed measurements should be capped at a lower limit of around 0.4 m/s due to limitations of the internal bearings that cause inherent noise in the data.

Response: The reported values are the 11-water year average wind speed for wind-sheltered stations. The wind speeds were sampled at 10 s intervals, then averaged to 15 min. and these averaged values were then averaged over 11 years. As mountain winds are gusty, any averaged wind speed measurement will contain information collected both above and below the stall speed of the anemometer. It would bias the wind speed to cap the anemometer recordings at 0.4 m/s. For instance, the value of 0.1 m/s is from the Upper Forest station that is situated in a mature spruce forest, a very calm and sheltered site from wind. From many field visits, the RM propeller was observed to be not turning, during which many 10 s measurements of 0 m/s were recorded.

pg. 6, line 11-15 - When calculating the mean incoming solar radiation, are nighttime hours included?

Response: Yes, when calculating these 11-water year average values, the nighttime hours are included. We were just trying to provide some general descriptive information such as these 11-water year averages to readers. We did not include more complex analyses beyond that for this data paper that is used to describe data collection method, data length and availability.

pg. 7, line 21-22 - The last sentence of this paragraph is unclear. When you say 'detailed survey data', are you just referring to occasional months with two measurements? The reader could also take that to mean the data include individual hole-by-hole SWE measurements, which is not provided here.

Response: Yes, by "detailed survey data", we mean snow survey data from measurements more than once per month. We added more description to this sentence to clarify that.

Figures 4–8 - Include some light gridlines in these plots make them easier to comprehend.

Response: Yes, we added light gridlines in Figures 4-8.

Figure 10 - The data in each of these plots are not from the same stations, so I suggest making them different marker types and including a legend.

Response: Yes, we used different marker types for data collected during historical and recent periods, and we also included a legend.

TECHNICAL CORRECTIONS

pg. 8, line 12 - Replace 'locations access challenges' with 'site access challenges'.

Response: Yes, we replaced 'locations access challenges' with 'site access challenges'.

pg. 11, line 7 - '...diagnose the basin response...'

Response: Yes, we corrected that.

Table 2 - It seems that there are problems with the reported AGS of the incoming solar radiation row for the Upper Clearing Tower and Vista View columns. For instance, below the 'Kipp and Zonen CM21 Pyranometer' there is a hanging '20', which looks like it should go below where you have an 'n/a'.

Response: For Upper Clearing Tower station, only incoming solar radiation and incoming longwave radiation were measured at 20 m. So, the two '20' below incoming solar radiation and incoming longwave radiation are just for these two components, and 'n/a' is for the not-measured outgoing components of solar and longwave radiation at 20 m. Similarly, for Vista View station, only incoming solar radiation was measured at 1.97 m, so the '1.97' below that is just for incoming solar radiation. The rest of 'n/a' are for not-measured radiation components. To make this more clear, in the variable name column, we added AGS (m) below Incoming Solar Radiation and AGS (m) below Incoming Longwave Radiation.